# Mid-Regional Pro-Adrenomedullin as a Prognostic Factor for Severe COVID-19 ARDS

**DOI:** 10.3390/antibiotics11091166

**Published:** 2022-08-29

**Authors:** Etienne de Montmollin, Katell Peoc’h, Mehdi Marzouk, Stéphane Ruckly, Paul-Henri Wicky, Juliette Patrier, Pierre Jaquet, Romain Sonneville, Lila Bouadma, Jean-François Timsit

**Affiliations:** 1Université de Paris Cité, INSERM, UMR 1137, IAME, 75018 Paris, France; 2Medical and Infectious Diseases Intensive Care Unit, APHP, Bichat-Claude Bernard Hospital, 75018 Paris, France; 3Université de Paris Cité, INSERM, UMRs 1149, CRI, 75018 Paris, France; 4Biochemistry Department, APHP, Bichat-Claude Bernard Hospital, 75018 Paris, France

**Keywords:** Mid-regional Pro-adrenomedullin, COVID-19, SARS-CoV-2, ARDS, biomarker, prognosis, superinfection

## Abstract

Mid-regional proadrenomedullin (MR-proADM) protects against endothelial permeability and has been associated with prognosis in bacterial sepsis. As endothelial dysfunction is central in the pathophysiology of severe SARS-CoV-2 infection, we sought to evaluate MR-proADM both as a prognostic biomarker and as a marker of bacterial superinfection. Consecutive patients admitted to the ICU for severe SARS-CoV-2 pneumonia were prospectively included and serum was bio-banked on days 1, 3, and 7. MR-proADM levels were measured blindly from clinical outcomes in batches at the end of follow-up. Among the 135 patients included between April 2020 and May 2021, 46 (34.1%) had died at day 60. MR-proADM levels on days 1, 3, and 7 were significantly higher in day-60 non-survivors. The area under the curve (AUC) of the receiver operating characteristic (ROC) curve (0.744, *p* < 0.001) of day-1 MR-proADM compared favorably with the AUC ROC curve of day-1 procalcitonin (0.691, *p* < 0.001). Serial MR-proADM measurements on days 3 and 7 may add prognostic information. After adjusting for CRP, LDH, and lymphocyte values, day-1 MR-proADM remained significantly associated with day-60 mortality. MR-proADM concentrations were significantly higher in patients with respiratory superinfections (on days 3 and 7) and bloodstream infections (on days 1, 3, and 7) than in patients without infection. Our results suggest that MR-proADM is a good predictor of outcome in severe SARS-CoV-2 infection and could be a useful tool to assess bacterial superinfection in COVID-19 patients.

## 1. Introduction

Coronavirus Disease (COVID-19) can present with a wide range of clinical severity, from asymptomatic infection to acute respiratory distress syndrome (ARDS) and death. As of August 2022, 591.7 million cumulative cases and 6.5 million deaths due to COVID-19 have been reported to the World Health Organization [1]. Acute hypoxemic respiratory failure is the main reason for intensive care unit (ICU) admission, and critically ill COVID-19 patients show mortality rates up to 40% [2]. The progression from a mild to severe disease is multifactorial but appears mainly driven by significant inflammation and microvascular thrombosis with evidence of endotheliitis [3,4,5].

Adrenomedullin is an endogenous vasoregulatory peptide that has been shown to play a role in preserving the integrity and stability of the endothelium after severe infection [6]. Mid-regional proadrenomedullin (MR-proADM) is used as a surrogate marker for adrenomedullin, as its levels are directly proportional to adrenomedullin which has a short half-life. Higher levels of MR-proADM in septic critically ill patients have been associated with disease severity [7] and mortality [8,9]. MR-proADM has also shown potential as a risk stratification biomarker for bacterial community-acquired pneumonia [10]. Finally, MR-proADM also appears to be a useful diagnostic tool for sepsis [11].

Being a marker of endothelial dysfunction, MR-proADM has been investigated in SARS-CoV-2 infection. Higher levels have been associated with mortality in the general population [12,13,14,15] and MR-proADM appeared to be a good risk stratification tool in the specific settings of the Emergency Room [16,17] and the ICU [18,19,20]. However, studies performed in the ICU setting used small patient samples with high heterogeneity of patient severity [21] and measured outcomes ≤ 30 days after ICU admission, which may be insufficient for such a study population.

Severe COVID-19 patients are at increased risk of bacterial superinfections, which contribute to ICU mortality [22,23]. These infections may be difficult to diagnose and lead to antibiotic overuse, which has led to a spread of antimicrobial resistance since the beginning of the pandemic [24]. In this context, MR-proADM could be an interesting biomarker for the diagnosis of bacterial superinfection. Surprisingly, no study evaluated MR-proADM as a diagnostic tool for bacterial sepsis in COVID-19 patients.

In this study, we aimed to evaluate MR-proADM as a prognostic biomarker in critically ill patients with severe SARS-CoV-2 pneumonia, and as a diagnostic tool for bacterial superinfection.

## 2. Results

### 2.1. Population Characteristics

Between April 2020 and May 2021, among 1294 admissions to the ICU of Bichat-Claude Bernard university hospital, 358 patients had severe SARS-CoV-2 pneumonia confirmed by polymerase chain reaction (PCR) and 135 patients with systematic bio-banking, and at least one measurement at day 1, 3, or 7 was included in the analysis (Appendix A) and followed-up for 39 (13–126) days. Day-1, day-3, and day-7 MR-proADM measurements were available in 120 out of 135 (88.9%), 119 out of 126 (94.4%), and 83 out of 86 (96.5%) patients, respectively. Among the three proADM measurement time points, 69 patients completed three, while 49 patients completed two, and 17 patients completed one.

Included patients were females in 43 (31.9%) cases, with a median age of 62.7 (51.6–71.2) years and an admission Simplified Acute Physiology Score (SAPS) II score of 27 (21–39) (Table 1). ICU admission occurred 9 (7–12) days after symptom onset. On day 1, 11 (8.1%) patients were under veno-venous extracorporeal membrane oxygenation, 19 (14.1%) patients were under invasive mechanical ventilation, 78 (57.8%) had non-invasive oxygenation techniques (non-invasive ventilation, continuous positive airway pressure or high flow nasal oxygen), and 27 (20%) had standard oxygen support. Antiviral treatment consisted of remdesivir for 65 (48.1%) patients. Immunomodulating treatments consisted of steroids in 127 (94.1%) cases, and anti-IL6 (tocilizumab) or anti-IL-1 (anakinra) in three (2.2%) cases. During ICU stay, 59 (44%) patients required invasive mechanical ventilation, 27 (20%) vasopressors, and 30 (22%) renal replacement therapy. ICU, hospital, and day-60 mortality rates were 30.4%, 36.3%, and 34.1%, respectively. ICU and hospital lengths of stay were 10 (6–22) and 16 (10–31) days, respectively.

### 2.2. Mid-Regional Pro-Adrenomedullin and Day-60 Mortality

Values of MR-proADM and several other biological markers are presented in Table 1. In patients that died before day 60, MR-proADM levels were significantly higher at all time points compared to survivors (Table 1 and Figure 1a). The areas under the receiver operating characteristic curve (AUROC) of MR-proADM for predicting day-60 mortality were 0.74 on day 1, 0.73 on day 3, and 0.74 on day 7 (Figure 1b). When choosing a cut-point of 1 nmol/L for day-1 MR-proADM (median value of the study population), sensitivity and specificity for predicting day-60 mortality were 77.5% (95% confidence interval (CI) 62.5–87.7) and 68.8% (95% CI 57.9–77.9), respectively.

Survival curves according to this cut-point are presented in Figure 2. Survival curves according to the same cut-point for day-3 and day-7 MR-proADM are presented in Appendix A and show a significant difference in day-60 mortality (log-rank test, *p* < 0.001 on day 3 and *p* = 0.002 on day 7). In a landmark analysis on day 3, the delta between day-3 and day-1 MR-proADM was significantly associated with day-60 mortality (HR 1.20, 95% CI 1.01–1.43, *p* = 0.04) (Appendix A). On day 7, the delta between day-7 and day-1 MR-proADM concentrations was not associated with day-60 mortality (HR 1.26, 95% CI 0.90–1.76, *p* = 0.16).

On day 1, MR-proADM compared favorably to other prognostic biomarkers identified in the literature, AUROC for procalcitonin, ferritin, d-dimers, and C-reactive protein being 0.69, 0.63, 0.60, and 0.55, respectively (Figure 3). It also compared favorably to the Sequential Organ Failure Assessment (SOFA) score on day 1 (AUROC 0.65). The combination of MR-proADM and procalcitonin improved only slightly prognostic accuracy, with an AUROC of 0.76. In multivariate analysis and at all time points, after adjusting for C-reactive protein, lactate dehydrogenase, and lymphocyte count, MR-proADM remained significantly associated with day-60 mortality (Table 2).

### 2.3. Mid-Regional Pro-Adrenomedullin and Bacterial Infections

During ICU stay, 52 (38.5%) patients presented with bacterial nosocomial pneumonia with a delay of 7 (4.5–9) days, of which 36 (26.7%) patients were ventilator-acquired pneumonia. Bacteremia occurred in 34 (25.2%) patients during the same period, with a delay of 9.5 (6–13) days. MR-proADM levels on day 3 (1.2 (0.8–2.1) vs. 0.9 (0.7–1.5), *p* < 0.01) and day 7 (1.2 (0.9–2.5) vs. 0.9 (0.6–1.2), *p* < 0.01) were significantly higher in patients with bacterial pneumonia, but not on day 1 (1.1 (0.7–1.6) vs. 0.8 (0.7–1.5), *p* = 0.22). MR-proADM levels were significantly higher in patients with bacteremia on day 1 (1.3 (1.0–2.4) vs. 0.8 (0.7–1.3), *p* < 0.01), day 3 (1.4 (0.9–2.9) vs. 0.9 (0.7–1.5), *p* < 0.01) and day 7 (1.1 (0.9–2.6) vs. 0.9 (0.7–1.4), *p* < 0.01). MR-proADM levels according to the occurrence of bacteremia or bacterial pneumonia, on days 1, 3, and 7, are presented in Figure 4.

## 3. Discussion

### 3.1. Main Findings

Using high-quality prospectively collected data from critically ill COVID-19 patients admitted to a large French COVID-19 reference center, we showed that MR-proADM concentrations were strongly associated with day-60 mortality. The prognostic accuracy of baseline MR-proADM was higher than commonly measured laboratory parameters and the SOFA score. We also showed that the AUROC for the prediction of day-60 mortality remained high on days 3 and 7, but that serial measurements might not be useful at all time points. When choosing a cut-point of 1 nmol/L, sensitivity and specificity for predicting day-60 mortality were 77.5% and 68.8, respectively, with good discrimination of survival curves. When evaluating the predictive accuracy of respiratory bacterial superinfection, MR-proADM concentrations were not significantly higher at baseline.

### 3.2. Interpretation

Regarding mortality risk stratification, our results are in line with previous studies performed in the ICU population showing AUROCs between 0.73 and 0.85 for 28-day mortality, with optimal cut-points between 1 and 1.8 nmol/L [18,19,20]. We used cut-points according to the MR-proADM distribution and previous studies to avoid overfitting [25]. Thus, the value of 1 nmol/L we chose was close to the median value of our sample and is in the range of published cut-points for mortality. Comparatively, in the general ward [12,13,14,15] and the emergency room [16,17] settings, MR-proADM also showed interesting risk stratification capabilities, with risk prediction of ICU admission, need for invasive mechanical ventilation, or death. These results have been confirmed in a pooled analysis of 6 studies and 487 patients, where MR-proADM values were increased by 74% (95% CI 46–103) in COVID-19 patients with critical illness compared to those without [21]. We believe risk stratification to be of paramount importance for COVID-19 patients, as patients with the highest values may benefit most from anti-viral or anti-inflammatory therapies such as steroids, interleukin-6 receptor antagonists, or anti-JAK molecules. In the context of COVID-19 patients, we show that MR-proADM has better prognostic capabilities than procalcitonin. These results are in accordance with published literature regarding the general sepsis population, where MR-proADM appeared to be a prognostic biomarker superior to procalcitonin [26,27].

The analysis of serial measurements of MR-proADM on days 1, 3, and 7 brings valuable information. First, we showed that prognostic accuracy for day-60 mortality is equivalent at each time point, meaning that MR-proADM can be measured at any time during the first few days of ICU admission. Second, the delta between day-3 and day-1 MR-proADM was significantly associated with day-60 mortality, suggesting that MR-proADM could be used to monitor COVID-19 patients. The fact that the delta between day-7 and day-1 MR-proADM was not significantly associated with day-60 mortality may be related to a loss of power due to a smaller patient sample on day 7. In a general population of 89 COVID-19 patients, Gregoriano et al. also found that MR-proADM remained low during the whole follow-up period in survivors, whereas non-survivors had a step-wise increase from baseline [14]. Given the small patient sample, our results warrant further studies to evaluate MR-proADM as a monitoring biomarker of disease progression.

MR-proADM has been proven a performing biomarker for the diagnosis of sepsis, with a calculated AUROC of 0.91 in a recent meta-analysis, and an optimal cut-point value of 1–1.5 nmol/L [11]. It has also shown a good diagnostic accuracy in specific infections, such as complicated urinary tract infections [28] and spontaneous bacterial peritonitis [29]. Thus, we sought to evaluate the diagnostic performance of MR-proADM for bacterial superinfection. Interestingly, we found that MR-proADM concentrations were significantly higher in patients with bacteremia and bacterial pneumonia. Bacterial pneumonia is frequent in severe COVID-19 ARDS, with up to 44% of mechanically ventilated patients developing ventilator-acquired pneumonia [30]. These infections can be difficult to diagnose, as COVID-19 patients may exert persistent systemic inflammation, and chest x-rays may be difficult to analyze due to the underlying viral pneumonia. While at the time of intubation less than 25% of patients present bacterial superinfection, ICU patients are frequently given systematic empiric antibiotic therapy. This strategy has led to an increase in antimicrobial resistance [24], and tools to identify patients with a high probability of bacterial superinfection are dearly needed. As such, we show that MR-proADM could be a useful marker to monitor bacterial superinfection in these patients, but our results need validation in larger cohorts.

### 3.3. Strenghts and Limitations

The strengths of our study are the prospective design and quality of collected data, including a follow up of 60 days, relevant for severe COVID-19 patients with extensive lengths of stay. The limitations of our study are: (1) a monocentric study design, (2) the small patient sample, despite being the largest published cohort in the literature, (3) the lack of external validation, and (4) the span of the study, covering different COVID-19 waves and SARS-CoV2 variants. Indeed, during each wave, the dominant variant presented distinct clinical and biological characteristics, including inflammatory response profile [31,32]. Hence, the prognostic accuracy of MR-proADM for each variant may have differed, but this has not been evaluated due to an insufficient patient sample.

## 4. Materials and Methods

### 4.1. Study Population

From April 2020 to May 2021, we included all adult patients that were admitted to the medical ICU of our hospital for severe SARS-CoV2 pneumonia and had had prospective serum bio-banking in the context of the OUTCOMEREA database. The bio-banking was conducted with the understanding and consent of each participant or surrogate. The OUTCOMEREA database has been approved by the French Advisory Committee for Data Processing in Health Research and the French Informatics and Liberty Commission (CNIL, registration no. 8999262). The database protocol was submitted to the Institutional Review Board of the French society of intensive care (CE-SRLF 22-76) on 12 September 2021. There were no exclusion criteria.

### 4.2. Data Collection and Definitions

The primary endpoint was the survival rate 60 days after ICU admission, and patients were followed-up to this time point or death. Data were prospectively collected at admission (demographics, chronic diseases, admission features, baseline severity indexes) and daily throughout the ICU stay (specific SARS-CoV2 treatments, need for invasive mechanical ventilation, need for vasopressors, need for renal replacement therapy, bacterial pneumonia and bacteremia, length of stay (LOS) and vital status at ICU and hospital discharge), using an anonymized electronic case report form. Severity of illness was graded at ICU admission with the use of the SAPS II [33] and the SOFA scores [34]. Immunodepression was defined as the use of long-term (>3 months) steroids, use of other immunosuppressant drugs, solid organ transplantation, solid tumor requiring chemotherapy in the last 5 years, hematologic malignancy, or HIV infection.

Serums were systematically collected on days 1, 3, and 7 (unless the patient was discharged from the ICU), allowed to clot at room temperature for 45 min, and then aliquoted and stored at −80 °C until assayed. Day-1, day-3, and day-7 MR-proADM concentrations were then measured blindly from clinical outcomes in batches at the end of follow-up, using an immunological assay with the TRACE technology (B·R·A·H·M·S MR-proADM KRYPTOR assay).

### 4.3. Statistical Analysis

Quantitative variables are presented as median, 1st, and 3rd quartiles, and compared between groups with the Mann–Whitney test or t-test, as appropriate. Qualitative variables are presented as frequency and percentage and compared with the Chi-square test or Fisher exact test as appropriate. For the analysis of bacterial superinfections, only episodes occurring after each time point analysis were considered. The association of MR-proADM with day-60 mortality was determined using a Cox proportional hazard model, adjusted on biomarkers associated with the outcome in the literature. Landmark analysis was used to evaluate this association at each time point (day 1, day 3, and day 7). Missing data, when at random, were handled by multiple imputations [35]. All statistical analyses were carried out with SAS 9.4 (SAS Institute Inc., Cary, NC, USA). A *p*-value of 0.05 and lower was considered statistically significant.

## 5. Conclusions

Our results suggest that MR-proADM is a promising predictor of outcome in critically ill patients with severe SARS-CoV-2 infection, superior to procalcitonin or the SOFA score. Furthermore, serial measurements may help monitor disease progression. In the event of future COVID-19 waves, MR-proADM could be used both for risk stratification and triage of patients presenting with severe SARS-CoV-2 pneumonia and for monitoring of bacterial superinfection in COVID-19 ICU patients.

## Figures and Tables

**Figure 1 antibiotics-11-01166-f001:**
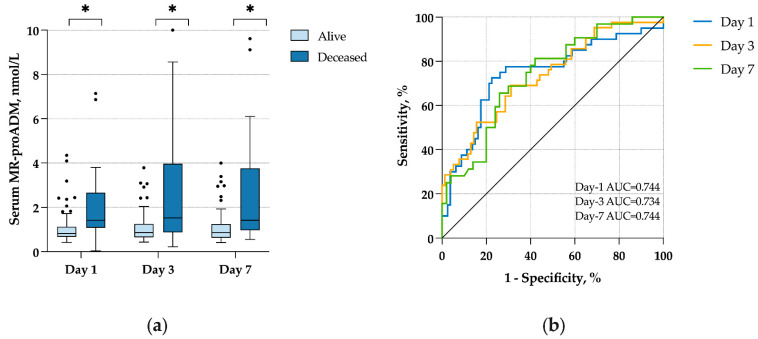
(**a**) MR-proADM concentrations according to day-60 survival; (**b**) ROC curves for day-60 survival of MR-proADM on days 1, 3, and 7. * *p* < 0.05.

**Figure 2 antibiotics-11-01166-f002:**
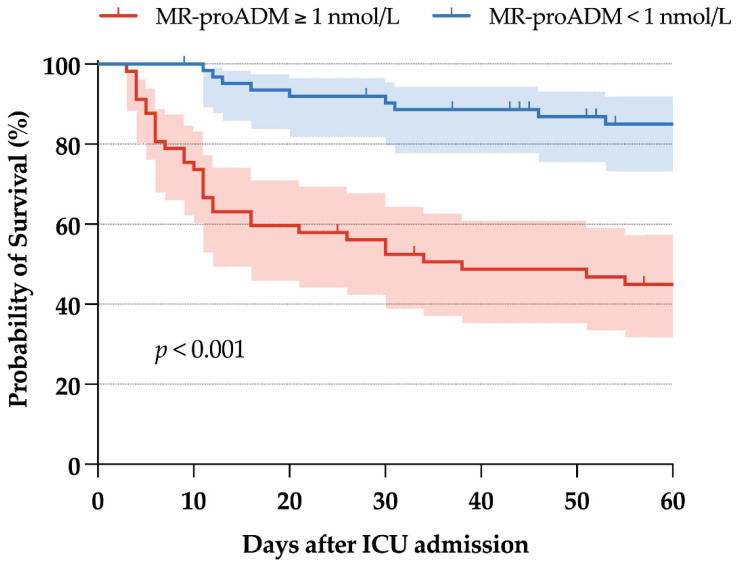
Survival curves according to a cut-point of 1 nmol/L of MR-proADM on day 1.

**Figure 3 antibiotics-11-01166-f003:**
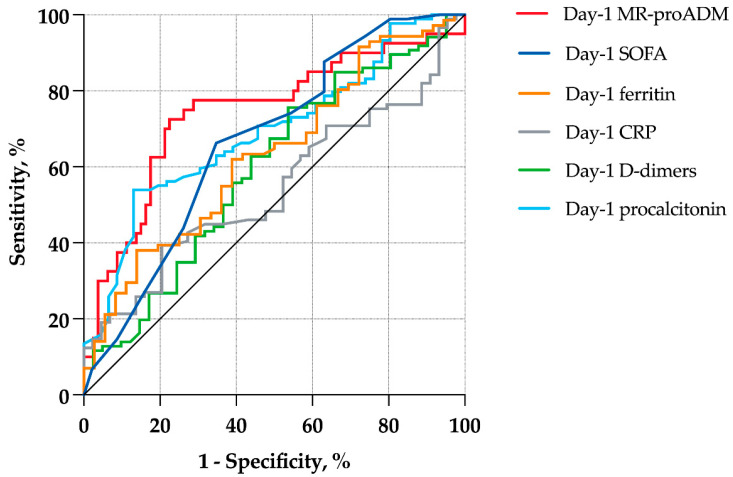
ROC curves of the association of day-60 mortality with various biomarkers on day 1 and the SOFA score. Abbreviations: MR-proADM, Mid-regional proadrenomedullin; SOFA, Sequential Organ Failure Assessment; CRP, C-reactive protein.

**Figure 4 antibiotics-11-01166-f004:**
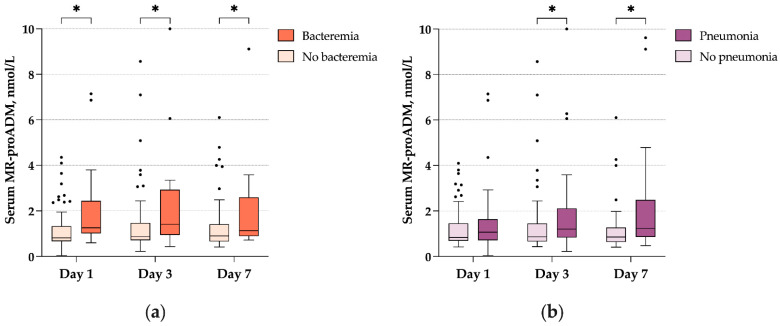
(**a**) MR-proADM concentrations according to the occurrence of pneumonia on days 1, 3, and 7. (**b**) MR-proADM concentrations according to the occurrence of pneumonia on days 1, 3, and 7. Abbreviations: MR-proADM, Mid-regional proadrenomedullin. * *p* < 0.05.

**Table 1 antibiotics-11-01166-t001:** Population characteristics at ICU admission.

	All Patients n = 135	Day-60 Survivors n = 89	Day-60 Decedents n = 46	*p*
**Demographics and comorbidities**				
Age, years	62.7 (51.6–71.2)	58.3 (49.2–66.1)	71.1 (62.3–75.9)	<0.01
Female gender	43 (31.9)	30 (33.7)	13 (28.3)	0.52
Body Mass Index	29 (25.8–34)	29.6 (26.3–34)	28.6 (24.3–31.2)	0.09
Diabetes	40 (29.6)	17 (19.1)	23 (50)	<0.01
Chronic diseases (Knaus ≥ 1)	60 (44.4)	36 (40.4)	24 (52.2)	0.19
Immunodepression	16 (11.9)	5 (5.6)	11 (23.9)	<0.01
**Characteristics at ICU admission**				
SAPS II	27 (21–39)	25 (18–34)	34 (26–48)	<0.01
Time from 1st symptoms to ICU admission, days	9 (7–12)	9 (7–12)	8 (7–15)	0.80
Respiratory SOFA	3 (1–4)	3 (1–4)	3 (2–4)	0.50
Extra-respiratory SOFA	1 (0–4)	1 (0–3)	1 (1–6)	<0.01
Ventilatory status at day 1				0.25
None	27 (20)	19 (21.3)	8 (17.4)	.
NIV/HFNC/CPAP	78 (57.8)	54 (60.7)	24 (52.2)	.
IMV/ECMO	30 (22.2)	16 (18)	14 (30.4)	.
Steroid therapy at day 1	127 (94.1)	85 (95.5)	42 (91.3)	0.44
**Laboratory data**				
MR-proADM, nmol/L				
Day 1	1 (0.7–1.6)	0.8 (0.7–1.1)	1.4 (1.1–2.7)	<0.01
Day 3	0.9 (0.7–1.7)	0.9 (0.7–1.2)	1.5 (0.9–3.6)	<0.01
Day 7	1 (0.7–1.9)	0.9 (0.6–1.2)	1.4 (1–3.6)	<0.01
IL-6 at day 1, pg/mL	36.3 (9–88)	29.4 (7.6–72)	58.3 (13.2–168)	0.11
CRP at day 1, mg/L	127 (68–177)	129 (57–177)	123.5 (93–172)	0.40
Procalcitonin at day 1, μg/L	0.3 (0.1–1.4)	0.2 (0.1–1)	0.6 (0.3–2)	<0.01
LDH at day 1, UI/L	439 (337–573)	427 (335.5–550)	474 (362–632)	0.16
Lymphocytes at day 1, G/L	0.97 (0.59–1.32)	1.01 (0.63–1.39)	0.85 (0.46–1.12)	0.04
D-dimers at day 1, μg/L	916 (556–1768)	811 (537–1431)	1161 (667–2594)	0.07
Ferritin at day 1, μg/L	851 (397–1668)	804 (344–1448)	1088 (575–2418)	0.03

Abbreviations: ICU, Intensive care Unit; SAPS, Simplified Acute Physiology Score; SOFA, Sequential Organ Failure Assessment; NIV, Non-invasive ventilation; HFNC, High Flow Nasal Canula; CPAP, Continuous Positive Airway Pressure; IMV, Invasive Mechanical Ventilation; ECMO, Extra-Corporeal Membrane Oxygenation, MR-proADM, Mid-regional proadrenomedullin; CRP, C-reactive protein; LDH, Lactate Dehydrogenase.

**Table 2 antibiotics-11-01166-t002:** Adjusted landmark analysis of the association of MR-proADM with day-60 mortality on days 1, 3, and 7.

	Hazard Ratio	95% Confidence Interval	*p*
**Landmark at day 1 (n = 135)**			
Day-1 MR-proADM	1.17	(1.06–1.28)	<0.01
Day-1 CRP	1.00	(1–1)	0.50
Day-1 lymphocytes count	1.00	(1–1)	0.28
Day-1 LDH	1.00	(1–1)	<0.01
**Landmark at day 3 (n = 135)**			
Day-3 MR-proADM	1.17	(1.06–1.28)	<0.01
Day-1 CRP	1.00	(0.99–1)	0.50
Day-1 lymphocytes count	1.00	(1–1)	0.28
Day-1 LDH	1.00	(1–1)	<0.01
**Landmark at day 7 (n = 128)**			
Day-7 MR-proADM	1.19	(1.1–1.31)	<0.01
Day-1 CRP	1.00	(0.99–1)	0.65
Day-1 lymphocytes count	1.00	(1–1)	0.62
Day-1 LDH	1.00	(1–1)	<0.01

Missing data imputed by multiple imputation. Hazard ratios are computed per one point of each variable. Abbreviations: MR-proADM, Mid-regional proadrenomedullin; CRP, C-reactive protein; LDH, Lactate Dehydrogenase.

## Data Availability

The datasets used and analyzed during the current study are available from the corresponding author on reasonable request.

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
