# Peer review of "Mid-Regional Pro-Adrenomedullin as a Prognostic Factor for Severe COVID-19 ARDS"

_antibiotics, 2022, doi:10.3390/antibiotics11091166_

Round 1

Reviewer 1 Report

I ask the authors to put the discussion behind the resultsI ask the authors to put the discussion behind the results.

Title

Mid-regional Pro-adrenomedullin as a prognostic factor for severe COVID-19 ARDS

Authors

Etienne De Montmollin , Katell Peoc’h , Mehdi Marzouk , Stéphane Ruckly , Paul-Henri Wicky , Juliette Patrier , Pierre Jaquet , Romain Sonneville , Lila Bouadma , Jean-François Timsit *

The topic of the article is Pro-adrenomedullin as a prognostic factor for severe COVID-19 ARDS

A very innovative parameter, which adds a new perspective on the severity of ARDS in COVID 19

Other parameters deal with the inflammatory response, this is a novelty in the literature

The topic is very original, and I would not be able to emphasize that it has been a long time since I have read such an original work, methodologically correct and concise.

Only English editing could do it

References are properly cited, adequately accompanying the manuscript itself

Frafically well received

I can't say bad..when it's good

Accept for print

Author Response

  • Reviewer 1
  • I ask the authors to put the discussion behind the resultsI ask the authors to put the discussion behind the results.”:
  • the “discussion” section follows the “results” section.
  • Only English editing could do it”:
  • English proofreading was performed on the revised manuscript.

Reviewer 2 Report

In this study, the authors showed that MR-proADM concentrations were strongly associated with day-60 mortality. Prognostic accuracy of baseline MR-proADM was also shown to be higher than commonly measured laboratory parameters.

Introduction:

1.     The introduction is very clearly written.

Methods:

2.     It was not clear what covariates were adjusted in the models.

3.     I would suggest the authors to account for diabetes status in the model given that people with diabetes may results in more severe complications.

Discussion:

4.     The impact of the virus may vary by variants given that the dominant variant changed across the study period. I would appreciate if the authors could discuss how this will bias the results.

5.     Did you calculate the power for this analysis? If there is enough power, the sample size is sufficient.

Author Response

  • English language and style are fine/minor spell check required”:
  • English proofreading was performed on the revised manuscript.
  • Methods: It was not clear what covariates were adjusted in the models” and “Methods: I would suggest the authors to account for diabetes status in the model given that people with diabetes may results in more severe complications”:
  • As we positioned our study on the biomarker level, we chose to adjust our multivariable model only on other biomarkers that had shown an association with prognosis in previously published literature. This is the reason we did not perform a multivariable model with clinical variables (and diabetes in particular).
  • Discussion: The impact of the virus may vary by variants given that the dominant variant changed across the study period. I would appreciate if the authors could discuss how this will bias the results.”:
  • we thank the reviewer for his sound comment. We have added this element in the limitation section and added 2 references.
  • Discussion: Did you calculate the power for this analysis? If there is enough power, the sample size is sufficient.”: As our main objective was to evaluate the prognostic accuracy of MR-proADM and we did not have any a priori hypothesis, we did not calculate any power. We believe it is safe to state that 135 patients is a small patient sample, and that our results must be reproduced in larger cohorts before advocating the use of MR-proADM in routine clinical practice.

Reviewer 3 Report

Authors wrote an interesting paper, well presented and with good scientific sound

Below only minor suggestions

1. Introduction: updata on SARS CoV2 cases wordwilde at the day of resubmission. Furthermore add also the role of bacteria infections during ARDS and the spread of AMR (antimicrobial resistance) during covid pandemic also in ICU unit as showed in this paper Impact of SARS-CoV-2 Epidemic on Antimicrobial Resistance: A Literature Review. Viruses. 2021 Oct 20;13(11):2110. doi: 10.3390/v13112110.

2. Method and result: I appreciate a lot the presentation and the quality of paper. Congratulations

3. Discussion: add information on MR-pro ADM as biomarker in sepsis and discuss also the role pf PCT and peak PCR. Discuss also the role of MDR bacteria in pneumonia in ICU (see also the paper suggested in introduction)

4. Conclision : give some proposal that came form your interesting data

5. Limitation: are honest and coherent with paper

Author Response

  • Reviewer 3
  • Introduction: update data on SARS CoV2 cases worldwide at the day of resubmission.”:
  • This information has been added in the introduction, as well as the link to the WHO dashboard in the references.
  • “Introduction: Furthermore, add also the role of bacterial infections during ARDS and the spread of AMR (antimicrobial resistance) during covid pandemic also in ICU unit as showed in this paper Impact of SARS-CoV-2 Epidemic on Antimicrobial Resistance: A Literature Review. Viruses. 2021 Oct 20;13(11):2110. doi: 10.3390/v13112110.”:
  • the introduction was expanded with a new paragraph, including the aforementioned reference.
  • Discussion: add information on MR-pro ADM as biomarker in sepsis and discuss also the role of PCT and peak PCR. Discuss also the role of MDR bacteria in pneumonia in ICU (see also the paper suggested in introduction)”:
  • We added AUC and optimal cut-points for the diagnosis of MR-proADM in sepsis from a recent meta-analysis, and discussed further the prognostic performance of PCT vs MR-proADM in sepsis.
  • Conclusion: give some proposal that came from your interesting data”:
  • In that respect, we have added a sentence in the conclusion.

Round 2

Reviewer 2 Report

Thank you for addressing my comments.